# Inhibition of TGF-β Signaling in Gliomas by the Flavonoid Diosmetin Isolated from *Dracocephalum peregrinum* L.

**DOI:** 10.3390/molecules25010192

**Published:** 2020-01-02

**Authors:** Yuli Yan, Xingyu Liu, Jie Gao, Yin Wu, Yuxin Li

**Affiliations:** 1National Engineering Laboratory for Druggable Gene and Protein Screening, Northeast Normal University, Jilin 130107, China; yanyl721@nenu.edu.cn (Y.Y.); liuxy013@nenu.edu.cn (X.L.); 2School of pharmacy, Changchun University of Chinese Medicine, Jilin 130107, China; 3Research Center of Agriculture and Medicine gene Engineering of Ministry of Education, Northeast Normal University, Jilin 130107, China; gaoj813@nenu.edu.cn

**Keywords:** *Dracocephalum peregrinum*, diosmetin, glioma cells, TGF-β, E-cadherin

## Abstract

**Background:***Dracocephalum peregrinum* L., a traditional Kazakh medicine, has good expectorant, anti-cough, and to some degree, anti-asthmatic effects. Diosmetin (3′,5,7-trihydroxy-4′-methoxyflavone), a natural flavonoid found in traditional Chinese herbs, is the main flavonoid in *D. peregrinum* L. and has been used in various medicinal products because of its anticancer, antimicrobial, antioxidant, estrogenic, and anti-inflammatory effects. The present study aimed to investigate the effects of diosmetin on the proliferation, invasion, and migration of glioma cells, as well as the possible underlying mechanisms. **Methods:** 3-(4,5-dimethylthiazol-2-yl)-2,5-diphenyltetrazolium bromide (MTT), scratch wound, and Transwell assays were used to demonstrate the effects of diosmetin in glioma. Protein levels of Bcl-2, Bax, cleaved caspase-3, transforming growth factor-β (TGF-β), E-cadherin, and phosphorylated and unphosphorylated smad2 and smad3 were determined by Western blots. U251 glioma cell development and progression were measured in vivo in a mouse model. **Results:** Diosmetin inhibited U251 cell proliferation, migration, and invasion in vitro, the TGF-β signaling pathway, and Bcl-2 expression. In contrast, there was a significant increase in E-cadherin, Bax, and cleaved caspase-3 expression. Furthermore, it effectively reduced the tumorigenicity of glioma cells and promoted apoptosis in vivo. **Conclusion:** The results of this study suggest that diosmetin suppresses the growth of glioma cells in vitro and in vivo, possibly by activating E-cadherin expression and inhibiting the TGF-β signaling pathway.

## 1. Introduction

Gliomas, the most common primary and lethal type of tumor in the central nervous system (WHO grade IV), are an aggressive form of brain cancer with a 5-year median survival rate of less than 5% [1,2]. They are characterized by their rapid infiltrating growth, migration ability, and invasive tumor growth [3,4]. Moreover, glioma cells show proliferation, diffuse infiltration, high resistance to apoptosis, degradation of the extracellular matrix, and stimulation of cell migratory signaling pathways, thereby facilitating the invasion of healthy brain tissue [5,6]. The treatment for gliomas typically involves surgical resection and a combination of chemotherapy and radiotherapy. Despite such treatments, the effect on overall survival is minimal [7,8]. Therefore, there is clearly a pressing clinical need to develop novel therapeutic strategies to improve the outcomes for gliomas, identify critical carcinogenic pathways, and discover new therapeutic targets for gliomas. In this regard, the flavonoid diosmetin, isolated from *Dracocephalum peregrinum*, is currently under investigation. Furthermore, on the basis of recent technological advances, numerous Chinese medicines that affect the development of gliomas in vivo and in vitro have been identified [9,10,11,12], providing new research clues for the diagnosis and treatment of this type of cancer.

*D. peregrinum* is a herb belonging to the Labiatae family. It is widely distributed in the northern territory of China and is a medicinal herb that has been used in traditional Kazakh medicine to treat cold and liver diseases. *D. peregrinum* is known as “*Tekanbasjelanbas*” and “*tikanbasgarambas*” in the Kazakh language and has been documented as Kazakhstani medicine in records such as the Kazak Journal of Medicine, Flora of Chinese, and Flora of Xinjiang.

Diosmetin is an *O*-methylated flavone (3′,5,7-trihydroxy-4′-methoxyflavone) (Figure 1A) that is abundant in olive leaves (*Olea europaea* L.), citrus fruits (and in related products, e.g., lemon or bergamot juice), and extracts of some medicinal herbs [13,14]. Furthermore, the main flavonoid of *D. peregrinum*, diosmetin, exhibits a large variety of pharmacological activities, such as anti-inflammatory, antioxidative, anticancer, and antiremodeling effects in various disorders [15,16,17]. Furthermore, it can selectively induce apoptosis and inhibit the growth of cancer cells without affecting the growth of normal cells [18]. Previous studies have shown that diosmetin can ameliorate pancreatitis by inhibiting the generation of proinflammatory mediators such as interleukin (IL)-1β, IL-6, and tumor necrosis factor-α (TNF-α) [19]. Moreover, diosmetin has been shown to suppress the proliferation of various tumor cells, such as hepatic carcinoma, breast cancer, leucocythemia, and prostate cancer cells [13,20,21,22]. However, to the best of our knowledge, there has been no definitive report on the effect of diosmetin on glioma cells. Thus, in this study, we aimed to investigate the effects of diosmetin on glioma cells in vitro and in vivo and its ability to induce their apoptosis.

## 2. Results

### 2.1. Diosmetin Inhibits the Proliferation, Invasion, and Migration of Glioma Cells

MTT, cell scratch, and Transwell assays were carried out to investigate the effects of diosmetin on the proliferation, migration, and invasiveness of glioma cells. Cell proliferation was significantly reduced in cultures exposed to diosmetin (Figure 1B–D). More importantly, the viability of cells treated with 5, 10, 15, and 20 μg/mL diosmetin for 72 h was effectively reduced to 49.2%, 29.8%, 21.5%, and 18.3% that of untreated cells, respectively (inhibition of 50.8%, 70.2%, 76.5%, and 80.3%, respectively), in U251 cells; the inhibition was 56.3%, 62.5%, 73.6%, and 82.7%, respectively, in U138 cells, and 45.8%, 64.8%, 80.7%, and 93.1%, respectively, in T98 cells. The U251 cell line was selected for further experiments. The scratch-healing rate, an indicator of cell migration, of the diosmetin-treated cell group was significantly lower than that of the control group at both 12 and 24 h (Figure 2A–C; *p* < 0.01, *p* < 0.01). Furthermore, the invasiveness of the cells was reduced significantly after treatment with 10 μg/mL (42.1 ± 6.74) and 20 μg/mL (38.5 ± 4.74) diosmetin compared with that of the control group (84.4 ± 8.62) (Figure 2D,E; *p* < 0.01, *p* < 0.001). In summary, diosmetin inhibited the proliferation, invasion, and migration of glioma cells.

### 2.2. Diosmetin Induces the Apoptosis of Glioma Cells

The results of flow cytometry showed that diosmetin significantly induced the apoptosis of U251 cells. The apoptosis ratio in the diosmetin 10 and 20 μg/mL groups was 0.98 and 2.54 times that in the control group, respectively (Figure 3A,B; *p* < 0.01, *p* < 0.001); thus, apoptosis in the diosmetin group was increased compared with that in the control group. To elucidate the effect of diosmetin on the U251 cells, the levels of Bcl-2, Bax, and cleaved caspase-3 in different treatment groups were determined. The result indicated that the expression of Bcl-2 in the diosmetin 20 μg/mL group was only 71% of that in the control group (Figure 3C,D; *p* < 0.001), while that of cleaved caspase-3 and Bax was 1.58 (Figure 3C,E; *p* < 0.01) and 1.63 (Figure 3C,F; *p* < 0.01) times that of the control group, which indicates that diosmetin can induce the apoptosis of U251 cells.

### 2.3. Diosmetin Leads to Inhibition of the TGF-β Signaling Pathway and Activation of E-Cadherin Expression in Glioma Cells

TGF-β signaling and E-cadherin play key roles in tumor cell progression. To assess the activation of E-cadherin, we investigated the molecular mechanism of diosmetin-regulated cell migration and invasiveness. Compared with the corresponding control group levels, the protein levels of TGF-β1, p-Smad2, and p-Smad3 decreased by 46.3%, 23.5%, and 28.6% in the diosmetin 10 μg/mL group, respectively (Figure 4A–C, *p* < 0.001, *p* < 0.05, *p* < 0.01), and 51.2%, 28.5%, and 41.3% in the diosmetin 20 μg/mL group, respectively (Figure 4A–C; *p* < 0.001, *p* < 0.01, *p* < 0.001). In contrast, E-cadherin expression increased by almost 50% in U251 cells due to 10 and 20 μg/mL of diosmetin treatment (Figure 4A,E–G, *p* < 0.01, *p* < 0.001). The results of these assays demonstrate that diosmetin increases the expression of E-cadherin and inhibits the TGF-β signaling pathway.

### 2.4. Diosmetin Inhibits Glioma Tumor Growth In Vivo

Using a mouse model of glioma, we investigated the effects of diosmetin on U251 glioma cells. In the control group, the mean tumor weight of 2.43 g and volume of 1.89 mm^3^ were significantly higher than those in the other groups, with the values in the diosmetin 20 μg/mL group being the lowest (mean weight of 1.39 g and volume of 1.08 mm^3^; Figure 5A–C). These results indicate that diosmetin can slow tumor growth, and this effect is dose-dependent.

## 3. Discussion

Glioma remains one of the most common malignancies worldwide, and the associated morbidity and mortality continue to increase yearly [1]. The median survival is generally less than 1 year from the time of diagnosis, and even under the most favorable situations, most patients die within 2 years [7]. Diosmetin has been previously demonstrated to induce cell apoptosis and act as an anticancer compound in several cancers [23,24,25,26]. However, the effects of diosmetin on glioma have not been elucidated, and whether diosmetin can affect glioma development and the underlying mechanisms remains unknown. In the present study, we researched the effects of diosmetin on U251 glioma cancer cells. We found that it inhibited the proliferation, migration, invasion, and cell apoptosis of U251 glioma cancer cells by regulating the TGF-β signaling pathway, and that it promoted apoptosis in glioma xenograft tumors in nude mice. These findings demonstrate that diosmetin inhibits the proliferation and promotes the apoptosis of U251 glioma cancer in vitro and in vivo, indicating that diosmetin may be a promising novel drug for glioma.

Cell proliferation, migration, invasion, and apoptosis play important roles in cell growth [27,28]. The levels of cleaved caspase-3, Bcl-2, and Bax in cells can directly regulate apoptosis [29]. Caspase-3, one of the most important proteins in the caspase family, plays a major role in the final phase of apoptosis. The ratio of Bax and Bcl-2, a pair of proteins that complement each other in the Bcl-2 family, plays an important role in the regulation of intrinsic apoptotic signaling [30,31]. The results of the present study show that diosmetin effectively promotes apoptosis, downregulates the levels of Bcl-2, and elevates the levels of Bax and cleaved caspase-3 in glioma cells.

The TGF-β superfamily of proteins, ubiquitous in cells, are widely involved in cell development and progression, and their excessive expression is found in a variety of tumors including lung, liver, breast, and ovarian cancers [32,33,34,35]. The Smad cytoplasmic proteins, a series of key proteins in the TGF-β signaling pathway, are present in both normal and transformed cells [36]. TGF-β signaling plays its role in regulating cell growth by binding its type I and type II receptors and inducing the phosphorylation of Smad2 and Smad3 [37]. It has been reported that the levels of TGF-β1 and phosphorylated Smads are upregulated in some malignant tumors [38,39]. In the present study, we showed that diosmetin inhibits TGF-β signaling in glioma cells by suppressing the expression of TGF-β and elevating the levels of the active forms of Smad proteins (i.e., p-Smad2 and p-Smad3). These findings are consistent with those of previous reports on other tumor types.

E-cadherin, a single-chain transmembrane glycoprotein, maintains the morphology and structural integrity of epithelial cells and plays an important regulatory role in the epithelial-mesenchymal transition (EMT) [40]. Reducing or eliminating the expression of E-cadherin can induce EMT, which affords tumor cells strong metastatic and invasive capabilities [41]. The expression level of E-cadherin in various tumors has been shown to be reduced [42], while the low expression level in the brain of patients with glioma is associated with an increased risk of EMT, which can lead to deterioration of the glioma [43]. Our experiments show that the expression of E-cadherin was increased in the diosmetin groups, suggesting that diosmetin increases the expression of E-cadherin through the TGF-β signaling pathway, thus inhibiting malignant metastasis and invasion of glioma cells.

## 4. Materials and Methods

### 4.1. Plant Material

In this study, the medicinal material used was the whole plant of *D. peregrinum* obtained from the Tori Region in northwestern Xinjiang in the autumn of 2014. The herb was stored at room temperature and identified by the Xinjiang Food and Drug Administration as belonging to the Labiatae family. Plant specimens were preserved in the National Engineering Laboratory for Drug Gene and Protein Screening in Northeast Normal University.

### 4.2. Isolation of Diosmetin from D. peregrinum

Air-dried *D. peregrinum* was cut into 3-cm-long pieces, and then refluxed three times with 70% ethyl alcohol. The resulting ethyl alcohol extract was successively fractionated with petroleum ether, chloroform, ethyl acetate, and n-butanol. The chloroform fraction was then separated by silica gel column chromatography with chloroform–methanol (100:0, 70:1, 50:1, 20:1, 5:1, 2:1, 0:100) to yield seven fractions. The yield was about 0.028% (chloroform fraction 80.0 g, diosmetin 25.0 mg). The effective fractions were purified using Sephadex-LH20 and analyzed by ^1^H NMR and ^13^C NMR (Table 1).

### 4.3. Cell Culture

The human glioma cell lines U251, U138, and T98 were obtained from the Shanghai Cell Bank, Type Culture Collection Committee, Chinese Academy of Sciences (Shanghai, China). The cells were cultured in Dulbecco’s modified Eagle’s medium (DMEM, Gibco, Invitrogen, Grand Island, NE, USA) containing 10% fetal bovine serum (FBS, HyClone, Logan, UT, USA), penicillin (100 U/mL), and streptomycin (100 μg/mL) in a humidified atmosphere containing 5% CO_2_ at 37 °C. Exponential phase cells were used in subsequent experiments.

### 4.4. MTT Assay

For the MTT assay, the cell suspension was diluted to a density of 2 × 10^3^ cells/well. Then, 200 µL of cell suspension was added to each well of 96-well plates and cultured for 24 h, before discarding the supernatant. Subsequently, MTT (Sigma-Aldrich, St. Louis, MO, USA) was added at a final concentration of 0.2 mg/mL, and the plates were incubated for 72 h. The medium was then replaced, and the cells were further cultured for 4 h at 37 °C in a humidified atmosphere with 5% CO_2_. The medium was removed, and then 200 µL of dimethyl sulfoxide (DMSO, Sigma-Aldrich, St. Louis, MO, USA) was added. The plates were gently mixed for 15 min to dissolve the formazan crystals completely. The optical density was measured at 490 nm using an ELX-800 (BioTek, Winooski, VT, USA).

### 4.5. Cell Scratch Assay

U251 cells in the exponential growth phase were mixed with serum-free medium. The cell suspension was diluted to a density of 1 × 10^5^ cells/well and inoculated in a six-well culture plate, with 200 μL of suspension per well. A marker pen was used to draw two parallel, equidistant lines along the bottom of the wells; the plate was placed in a 5% CO_2_ environment at 37 °C for 24 h to aid cell confluency. Then, a 200-μL micropipette was used to scratch a line evenly in the cell monolayer between the two marker lines. After the scratch, the medium was slowly aspirated using a pipette, and the cells were washed twice with serum-free medium to remove cell debris. The six-well plate was placed under an inverted microscope to observe and acquire images; the migration distance of the cells before and after incubation was measured, and the migration rate was calculated.

### 4.6. Transwell Assay

Cells in the exponential growth phase were digested with 0.25% trypsin without EDTA, and the diluted cell density was 2 × 10^4^ cells/well. The cells were resuspended with serum-free cell culture medium, and then seeded in the upper chamber of a Transwell apparatus precoated with Matrigel (Corning Incorporated Life Sciences, Tewksbury, MA, USA). The lower chamber was supplemented with 5% FBS. The cells were then cultured for 24 h in an incubator. The Transwell semipermeable membrane was removed, washed with PBS, fixed with paraformaldehyde for 20 min, and stained with 0.1% crystal violet for 30 min. To remove cells that did not migrate through the semipermeable membrane, the upper surface was wiped gently with cotton three times using PBS. Migratory cells on the bottom of the membrane were observed and counted (five random fields) under a 400× microscope, and the invasive ability of the cells was calculated as the number of cells penetrating the membrane.

### 4.7. Flow Cytometric Analysis of Apoptosis

Apoptotic cells were detected using the Annexin V-PI Apoptosis Detection Kit (KeyGEN BioTECH, Nanjing, China) and flow cytometry (FACSCalibur, BD, Franklin Lakes, NJ, USA) according to the manufacturer’s instructions. Cells were added to a six-well cell culture plate and incubated for 24 h. The medium was discarded, and the cell pellet was collected, washed twice with cold PBS, stained with 400 μL of 1× Annexin V and 5 μL of Annexin V-FITC, and incubated at 2–8 °C for 15 min in the dark. Subsequently, 10 μL of PI was added after staining by liquid blending and incubated at 2–8 °C in the dark for 15 min. Cell apoptosis was then analyzed by flow cytometry.

### 4.8. Western Blotting

The cells were lysed, collected, and cleared using the total protein extraction kit (Beyotime Institute of Biotechnology, Haimen, China). The extracts were then subjected to sodium dodecyl sulfate-polyacrylamide gel electrophoresis (SDS-PAGE) (10% separating gel, 4% stacking gel). The proteins were electro-transferred onto a polyvinylidene fluoride membrane and blocked with 5% non-fat milk at room temperature. The membrane was incubated with specific rabbit monoclonal primary antibodies (cleaved caspase-3, Bax, Bcl-2, TGF-β, Smad2, p-Smad2, Smad3, p-Smad3, E-cadherin, or β-actin, 1:1000; Bioss, Beijing, China) at 4 °C overnight and then incubated with horseradish peroxidase (HRP)-labeled goat anti-rabbit secondary antibody. The proteins were visualized using enhanced chemiluminescence (ECL) reagents (Qihai Biotec, Shanghai, China). Gel-Pro Analyzer software was used for imaging, and the signals were normalized using β-actin as the internal control.

### 4.9. Immunofluorescence Staining

The cultured cells were digested with 0.25% trypsin, fully dispersed into a single cell suspension, and grown on a glass slide. When the cells were close to confluence, the culture medium was discarded, and the cells were washed three times with PBS and fixed with 4% paraformaldehyde for 20 min. The glass slide was removed and washed twice with PBS, and then 0.1% Triton X-100 was added and incubated for 20 min. Thereafter, the cells were incubated with E-cadherin antibody (1:50; Beyotime) at 4 °C overnight. The washed cells were then incubated with Cy3-labeled goat anti-rabbit secondary antibody (Beyotime) for 1 h at room temperature and washed three times with PBS. After counter-staining the nuclei with DAPI, a laser confocal microscope (Olympus, Tokyo, Japan) was used to observe the images at 600× magnification.

### 4.10. Real-Time Polymerase Chain Reaction

Total RNA was prepared from cells using an RNA extraction kit (TIANGEN Biotech, Beijing, China). cDNA was then synthesized by reverse transcription. SYBR GREEN master mix (TIANGEN) was used for real-time PCR in an Exicycler™ 96 Real-Time Quantitative Thermal block (Bioneer, Daejeon, Korea), with β-actin as the internal control. The primers are listed in Table 2. The total qPCR reaction volume was 20 μL and consisted of 1 μL of cDNA, 0.5 μL of each primer, 10 μL of SYBR GREEN master mix, and 8 μL of ddH_2_O. The PCR reaction program was as follows: 95 °C for 10 min; 40 cycles of 95 °C for 10 s, 58 °C for 20 s, and 72 °C for 30 s; and 4 °C for 5 min.

### 4.11. In Vivo Experiments

In vivo experiments were conducted using pathogen-free BALB/c nude mice at 4–6 weeks of age, provided by the Beijing Vital River Laboratory Animal Technology Co. Ltd. (Beijing, China). The mice were housed in laboratory facilities at a temperature range of 21.0–25.0 °C, relative humidity of 40.0%–70.0%, and 12 h light/dark cycle; the air was exchanged at least 15 times per hour. The mice had free access to food and water. The care and treatment of the mice were approved by the Experimental Animal Ethics Committee of China Medical University. After one week of acclimatization, the mice were randomly assigned to the following four groups (n = 5 per group): U251, 5 μg/mL diosmetin (i.v.), 10 μg/mL diosmetin (i.v.), and 20 μg/mL diosmetin (i.v.). The cells (1 × 10^6^) were suspended in 0.2 mL of normal saline and then injected subcutaneously into the right side of the breast pad of the nude mice. Tumor volumes were calculated as follows: volumes = a × b^2^/2, where a is the larger of the two dimensions and b is the smaller. The mice were euthanized after 30 days and their tumors were weighed and photographed.

### 4.12. Data Analysis

Data are presented as means ± standard deviations. Comparisons between groups were performed using one-way analysis of variance, and multiple comparisons were performed using Bonferroni post-hoc tests. Data and figures were processed using GraphPad Prism 5.0 software. *p <* 0.05 was considered statistically significant.

## 5. Conclusions

Taken together, the results of the present study demonstrate that diosmetin inhibits the proliferation, invasion, and migration of glioma cells and increases the expression of the metastasis-related factor E-cadherin via the TGF-β signaling pathway. Our preliminary data suggest that diosmetin plays a key role in the malignant progression of gliomas and suggests that diosmetin can be further explored as a therapy drug for gliomas.

## Figures and Tables

**Figure 1 molecules-25-00192-f001:**
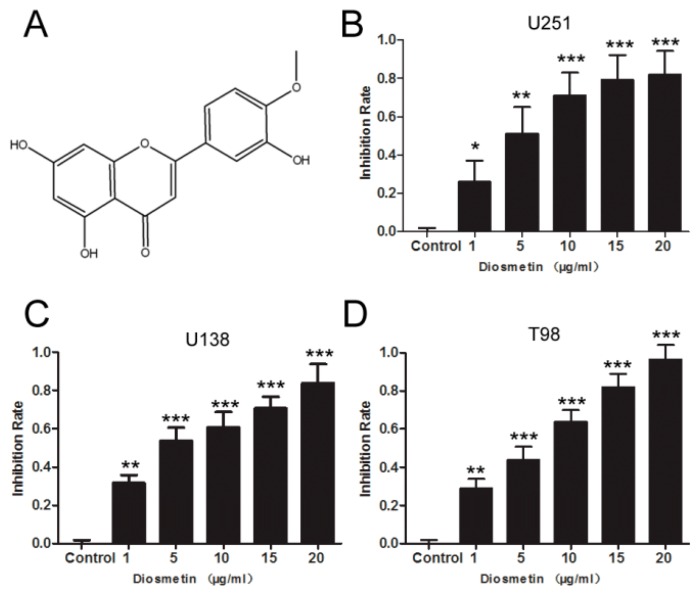
Diosmetin inhibited the proliferation of glioma cells. (**A**) The structure of diosmetin. MTT assay for the case of cell proliferation of (**B**) U251, (**C**) U183, and (**D**) T98; *n* = 6, and the data are presented as the mean ± standard deviation. Compared with the control group, * *p* < 0.05, ** *p* < 0.01, *** *p* < 0.001.

**Figure 2 molecules-25-00192-f002:**
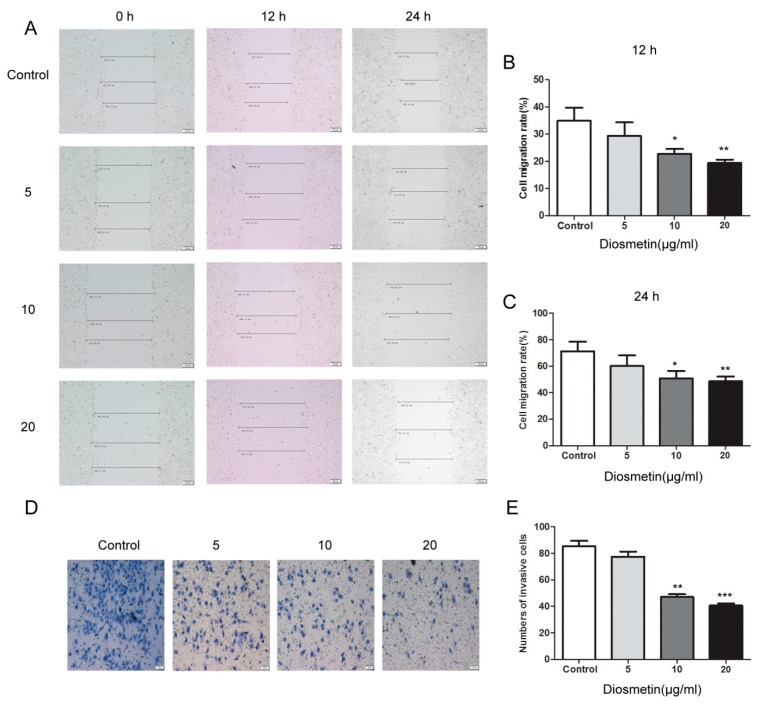
Diosmetin inhibited the invasion and migration of glioma cells. (**A**–**C**) Cell-scratching assays. (**D**,**E**) Transwell assays. Compared with group, * *p* < 0.05, ** *p* < 0.01, *** *p* < 0.001.

**Figure 3 molecules-25-00192-f003:**
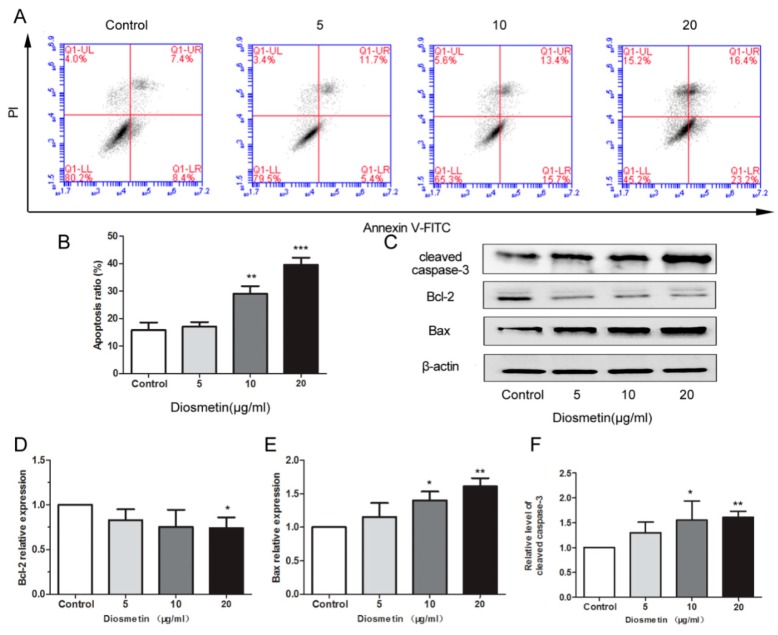
Diosmetin induced the apoptosis of glioma cells. (**A**,**B**) Flow cytometry analysis to examine cell apoptosis; (**C**–**F**) Western blot analysis was used to determine the expression of Bcl-2, Bax, cleaved caspase-3 in different groups. β-actin was used as an internal control for grayscale analysis. Compared with the control group, * *p* < 0.05, ** *p* < 0.01.

**Figure 4 molecules-25-00192-f004:**
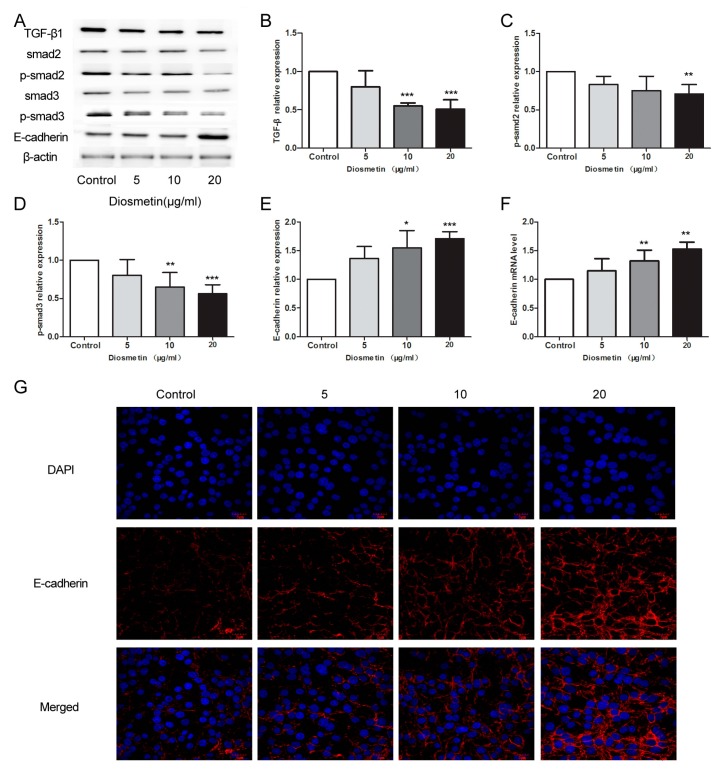
Diosmetin led to inhibition of the TGF-β signaling pathway and activation of E-cadherin expression in glioma cells. Western blot assay was performed to detect the protein levels of (**A**,**B**) TGF-β1, (**A**,**C**) p-smad2, (**A**,**D**) p-smad3 and (**A**,**E**) E-cadherin, with the grayscale analysis using β-actin as the internal control. (**F**) Real-time PCR was used to detect the expression level of E-cadherin mRNA. (**G**) Immunofluorescence staining was performed to observe the distribution of E-cadherin protein in glioma cells. Compared with control group, * *p* < 0.05, ** *p* < 0.01, *** *p* < 0.001.

**Figure 5 molecules-25-00192-f005:**
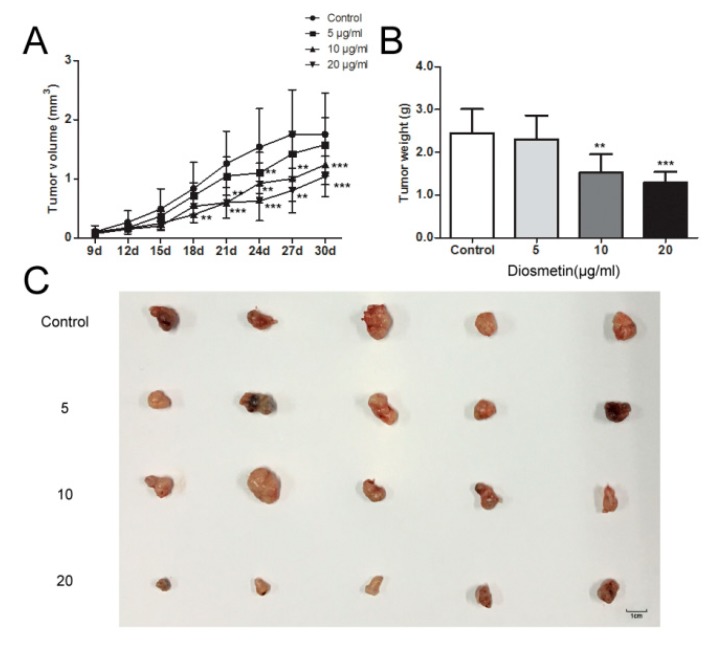
Diosmetin inhibited glioma tumor growth in vivo. (**A**) Tumor volume assay, (**B**) tumor weight assay, and (**C**) the photo of the tumor tissue. This figure shows the typical result of repeated experiments, and the data are presented as the mean ± standard deviation. Compared with the control group, ** *p* < 0.01, *** *p* < 0.001.

**Table 1 molecules-25-00192-t001:** ^1^H and ^13^C NMR spectral data of diosmetin observed at 400/100 MHz in DMSO-*d*_6_.

Pos.	^δ^H (mult., *J* in Hz)	^δ^C
2	-	163.50
3	6.73 (s)	103.72
4	-	181.66
5	12.91	157.29
6	6.18 (d, 2.0 Hz)	98.84
7	10.80	164.17
8	6.45(d, 1.6 Hz)	93.87
9	-	161.43
10	-	103.50
1′	-	118.68
2′	7.41 (d, 2.4 Hz)	112.93
3′	9.41	146.76
4′	-	151.11
5′	7.07 (1 H, d, 8.4 Hz)	112.14
6′	7.53 (1 H, dd, 2.0, 8.4 Hz)	122.98
-OCH_3_	3.85 (3 H, s)	55.74

**Table 2 molecules-25-00192-t002:** Sequences of the primers for real-time PCR.

Primer Name	Sequence (5′–3′)
E-cadherin F	ATGCCGCCATCGCTTACAC
E-cadherin R	CGACGTTAGCCTCGTTCTCA
β-actin F	CTTAGTTGCGTTACACCCTTTCTTG
β-actin R	CTGTCACCTTCACCGTTCCAGTTT

## Data Availability

The [DATA TYPE] data used to support the findings of this study are included within the article. Including: Graphpad Prism 5.0 software.

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
