# Peer review of "Inhibition of TGF-β Signaling in Gliomas by the Flavonoid Diosmetin Isolated from Dracocephalum peregrinum L."

_molecules, 2020, doi:10.3390/molecules25010192_

Round 1

Reviewer 1 Report

The work of Yan et al is quite interesting and results expand the actual scientific knowledge.

I wonder if autors tested the cytotoxicity of diosmetin in non-tumoral cells or if they can give any idea of the tumor-selectivity of diosmetin.

I suugest to express results as µM instead of µg/mL.

Author Response

Dear Editor,

Thank you for your careful guidance:

I have revised the manuscript one by one according to the guidance and adjusted the Figures.

For Reviewer 1

As diosmetin is one of the most important flavone in many traditional Chinese medicine. There has been widely reported that have no killing effect on ordinary cells. Such as the reference 15-17 reported. And also the results expressed as g/mL.

Reviewer 2 Report

Diosmetin has been shown to suppress the proliferation of various tumor cells, such as hepatic carcinoma, breast cancer, leucocythemia, and prostate cancer cells. This paper isolated diosmetin from Dracocephalum peregrinum L. and studied its effects on the proliferation, invasion, and migration of U251 glioma cells and the possible underlying mechanisms. The growth of diosmetin-treated U251 glioma cells injected subcutaneously into the right side of the breast pad of the pathogen-free BALB/c nude mice was also studied in vivo. The results suggested that diosmetin suppresses the growth of glioma cells in vitro and in vivo, possibly by activating E-cadherin expression and inhibiting the TGF-β signaling pathway. There are some concerns as listed in the following:

(1) In terms of inhibiting the TGF-β signaling pathway, one published paper, titled ‘Diosmetin prevents TGF-β1-induced epithelial-mesenchymal transition via ROS/MAPK signaling pathways’ (Life Sci. 2016 May 15;153:1-8), should be considered.

(2) It is unclear If the proapoptotic effect of diosmetin in the present study also is mediated via inhibiting the TGF-β signaling pathway. A recent paper reported anti-proliferation and pro-apoptotic effects of diosmetin via modulating cell cycle arrest and mitochondria-mediated Intrinsic apoptotic pathway in MDA-MB-231 cells. (Med Sci Monit. 2019 Jun 22;25:4639-4647)

(3) Please give the yield ratio in ‘2.2. Isolation of diosmetin from D. peregrinum’

(3) Typos and others

*L33: Abbreviations: listed in alphabetic order

L71: MTT assay the case of cell proliferation

L103: CO2

L104: added..

*L121: with ?% FBS

*L167: Add the method for ‘tumor volume assay’ as the data shown in Fig. 5A.

*L254-255: it inhibited the proliferation, migration, invasion, and? apoptosis of U251 glioma

*L273-274: by suppressing the expression of TGF-β and elevating? the levels of the active forms of Smad proteins

L276: cadherin

*L289-290: Our preliminary data suggest that diosmetin plays a key role? in malignant progression of glioma

L303: References: Keep one format for References, i.e. capital letter in title (R2, R8, R11 R18, R22), page number (R4, R18, R22, R23, R24, R26, R28)

*L360: R29: volume, page number?

Author Response

For Reviewer 2

(1) In terms of inhibiting the TGF-β signaling pathway, one published paper, titled ‘Diosmetin prevents TGF-β1-induced epithelial-mesenchymal transition via ROS/MAPK signaling pathways’ (Life Sci. 2016 May 15;153:1-8), should be considered.

(2) It is unclear If the proapoptotic effect of diosmetin in the present study also is mediated via inhibiting the TGF-β signaling pathway. A recent paper reported anti-proliferation and pro-apoptotic effects of diosmetin via modulating cell cycle arrest and mitochondria-mediated Intrinsic apoptotic pathway in MDA-MB-231 cells. (Med Sci Monit. 2019 Jun 22;25:4639-4647)

Response: I have read relevant articles, thank you for your recommendation.

(3) Please give the yield ratio in ‘2.2. Isolation of diosmetin from D. peregrinum’

Response: I have add this information in the manuscript as L87 show.

(4) Typos and others

*L33: Abbreviations: listed in alphabetic order

L71: MTT assay the case of cell proliferation

L103: CO2

L104: added..

*L121: with ?% FBS

*L167: Add the method for ‘tumor volume assay’ as the data shown in Fig. 5A.

*L254-255: it inhibited the proliferation, migration, invasion, and? apoptosis of U251 glioma

*L273-274: by suppressing the expression of TGF-β and elevating? the levels of the active forms of Smad proteins

L276: cadherin

*L289-290: Our preliminary data suggest that diosmetin plays a key role? in malignant progression of glioma

L303: References: Keep one format for References, i.e. capital letter in title (R2, R8, R11 R18, R22), page number (R4, R18, R22, R23, R24, R26, R28)

*L360: R29: volume, page number?

Response: I have change the information in the manuscript as the reviewer suggest.

Reviewer 3 Report

The manuscript entitled „Inhibition of TGF-β signaling in glioma by the flavonoid diosmetin isolated from Dracocephalum peregrinum L.” by Yan et al. is an interesting paper, which discusses a current topic, as the therapy of malignant CNS tumors is still an unresolved medical issue and with the advent of traditional Chinese medicinal plants, there is still many unanswered questions regarding the medicinal use of these plants. The paper is well-written, the methods used are current and appropriate, the results were presented appropriately and discussed accordingly.

I have the following remarks:

Reference numbers should be placed in brackets, based on the „Instructions for authors” section of the Journal.

Line 48-49: were these plants identified on the basis of evidence-based medicine (EBM) or just in vitro data? Please specify!

Line 50: Labiatae should be in italics.

Line 55-59: that means that diosmetin could have been isolated from those other plants in high concentrations as well? Why did the authors choose Dracocephalum peregrinum in that case? This is ambiguously written, please state it more clearly.

Terms like in vitro, in vivo should always be in italics.

I believe that Figure 1. should be in the Results section, not in the Introduction. Please correct this.

Line 91-96: no antifungal agent was used in the medium? If so, please specify why?

Line 98-105: Please add the following reference for Section 2.4. discussing the MTT assay:

https://www.sciencedirect.com/science/article/pii/S0960894X17300458

Why did the authors measure at 490nm, when in most cases 550-570nm measurements are used? Please discuss.

Line 129-136: Please add the following reference for Section 2.7. discussing the Annexin V-PI assay:

https://www.ncbi.nlm.nih.gov/pubmed/26026084

The authors should check the sections “Availability of data and materials”, „Patient consent for publication” and „Competing interests” for potential errors.

References are not according to the guidelines specified in the „Instructions for authors” section of the Journal, please correct.

The authors should do a language-check on the paper, especially when it comes to the use of commas.

I recommend publication after these minor revisions.

Author Response

Reference numbers should be placed in brackets, based on the „Instructions for authors” section of the Journal.

Response: I have modification the information in the manuscript as the reviewer suggest.

Line 48-49: were these plants identified on the basis of evidence-based medicine (EBM) or just in vitro data? Please specify!

Response: I have add the information in the manuscript as the reviewer suggest.

Line 50: Labiatae should be in italics.

Response: I have modification the information in the manuscript as the reviewer suggest.

Line 55-59: that means that diosmetin could have been isolated from those other plants in high concentrations as well? Why did the authors choose Dracocephalum peregrinum in that case? This is ambiguously written, please state it more clearly.

Response: Just as Dracocephalum peregrinum is widely distributed in the northern territory of China.

Terms like in vitro, in vivo should always be in italics.

Response: I have modification the information in the manuscript as the reviewer suggest.

I believe that Figure 1. should be in the Results section, not in the Introduction. Please correct this.

Response: I have modification the information in the manuscript as the reviewer suggest.

Line 91-96: no antifungal agent was used in the medium? If so, please specify why?

Response: The cultured cells in our laboratory can be kept sterile without adding antibacterial agents.

Line 98-105: Please add the following reference for Section 2.4. discussing the MTT assay:

https://www.sciencedirect.com/science/article/pii/S0960894X17300458

Why did the authors measure at 490nm, when in most cases 550-570nm measurements are used? Please discuss.

Line 129-136: Please add the following reference for Section 2.7. discussing the Annexin V-PI assay:

https://www.ncbi.nlm.nih.gov/pubmed/26026084

The authors should check the sections “Availability of data and materials”, „Patient consent for publication” and „Competing interests” for potential errors.

Response: Follow the instructions.

References are not according to the guidelines specified in the „Instructions for authors” section of the Journal, please correct.

The authors should do a language-check on the paper, especially when it comes to the use of commas.

Response: I have modification the information in the manuscript as the reviewer suggest.

Reviewer 4 Report

The manuscript by Yu-li Yan et al. try to provide evidence that diosmetin, the main flavonoid of D. peregrinum, exerts anti-cancer effects suppressing glioma growth in vitro and in vivo. However, there are important points that should be addressed prior to being considered for publication. Manuscript needs an overall improvement before a potential publication in any journals. My suggestion is to improve the manuscript and re-submit it again

Some suggestions are shown below.

-In general, the authors should better describe methods, including the description of how they managed row data to obtain the presented data (normalization, %, etc etc)

-MTT data shown are not clear, in particular the control ones, usually MTT data show cell viability, please show the raw OD data.

In addition, regarding the MTT method, authors should provide data also incubating cells with MTT salt for lower time; MTT is toxic for cells and they incubated cells with MTT for 72h, showing only one time point, presumably 72h. They should perform the same experiments incubating cells with MTT salt (0.5mg/ml) for only 2h at the end of treatments and they have to show also the effect on GBM viability at 24h (and 48h), that is the time point used for the scratch and transwell assay.

-Scratch assay data: the pictures showing wounded cell layers have too low resolution, it is impossible to read the measurements present (distances?), and, again, it is not possible to understand how authors obtained those %s (of what?). There are three lines, why? Again method section lacks of important information.

- Transwell assay: please better described the experiment: you used FBS as chemoattractant but at which concentration? Which magnification you used to count migrated cells?

In addition, without data of the effect of diosmetin on viability at 24h it is not possible to exclude that migration and invasion decreased because of the decrease of viability and consequently increased of apoptosis compared to control, as authors showed. Indeed, after 24h treatments authors showed an increase of 40% apoptosis compared to the basal 15% of control…

- Apoptosis data: please describe which quadrants you use for apoptosis analysis, probably you showed early + late apoptosis, that means the two right quadrants, but this information can not be found in the text, again.

-Immunofluorescence and wb data: again, it is impossible to find information about how immunofluorescence was quantified or the time of treatments for both assays. Because of the great effect of diosmetin on glioma viability authors must show these information

-please show the number of replicates: except for in vivo exps, authors never showed the N replicates per experiments, it’s incredible! Beyond the number of replicates, the figure legends lack of any basic information (treatment, time, concetration…)

-In vivo experiment is scarcely described, how many time authors treated the animals? How they deliver drugs? How they measured tumor volume along the time?

Without going any further, in my opinion this paper is to premature to be published.

Author Response

The manuscript by Yu-li Yan et al. try to provide evidence that diosmetin, the main flavonoid of D. peregrinum, exerts anti-cancer effects suppressing glioma growth in vitro and in vivo. However, there are important points that should be addressed prior to being considered for publication. Manuscript needs an overall improvement before a potential publication in any journals. My suggestion is to improve the manuscript and re-submit it again

Some suggestions are shown below.

-In general, the authors should better describe methods, including the description of how they managed row data to obtain the presented data (normalization, %, etc etc)

Response: In this paper, the form of data referred to many references such as reference 10 to 15, and combined with the original experimental data.

-MTT data shown are not clear, in particular the control ones, usually MTT data show cell viability, please show the raw OD data.

Response: In this paper, the data were calculated. The control group was not a blank group, but a control group with the solvent. And the calculation formula is mean OD value of the experimental group/mean OD value of the control group×100%.

In addition, regarding the MTT method, authors should provide data also incubating cells with MTT salt for lower time; MTT is toxic for cells and they incubated cells with MTT for 72h, showing only one time point, presumably 72h. They should perform the same experiments incubating cells with MTT salt (0.5mg/ml) for only 2h at the end of treatments and they have to show also the effect on GBM viability at 24h (and 48h), that is the time point used for the scratch and transwell assay.

Response: The experimental results are proportional to time, so the MTT results show 72 hours of data.

The experimental date of the scratch and transwell assay are experimental data that take into account the number of cells, experimental pictures and other factors.

-Scratch assay data: the pictures showing wounded cell layers have too low resolution, it is impossible to read the measurements present (distances?), and, again, it is not possible to understand how authors obtained those %s (of what?). There are three lines, why? Again method section lacks of important information.

Response: The experimental data chart itself has a resolution that meets the requirements, but has been reduced in the inserted article.

In order to clearly mark the distance of cell migration, the three lines obtained the average data through multiple experiments.

% shows cell mobility

- Transwell assay: please better described the experiment: you used FBS as chemoattractant but at which concentration? Which magnification you used to count migrated cells?

Response: I have add this information in the manuscript as L122 and L127 show.

In addition, without data of the effect of diosmetin on viability at 24h it is not possible to exclude that migration and invasion decreased because of the decrease of viability and consequently increased of apoptosis compared to control, as authors showed. Indeed, after 24h treatments authors showed an increase of 40% apoptosis compared to the basal 15% of control…

Response: This is an interesting question, but I believe that both inhibiting migration and promoting apoptosis inhibit the growth of tumor cells.

- Apoptosis data: please describe which quadrants you use for apoptosis analysis, probably you showed early + late apoptosis, that means the two right quadrants, but this information can not be found in the text, again.

Response: refer to many references the apoptosis date were showed early + late apoptosis as the paper shown.

-Immunofluorescence and wb data: again, it is impossible to find information about how immunofluorescence was quantified or the time of treatments for both assays. Because of the great effect of diosmetin on glioma viability authors must show these information

Response: The calculation method of this kind of experimental data is universal.

-please show the number of replicates: except for in vivo exps, authors never showed the N replicates per experiments, it’s incredible! Beyond the number of replicates, the figure legends lack of any basic information (treatment, time, concetration…)

Response: I don't think it's impossible to do multiple controls and multiple trials with cells, and that's what we do.

The problem you mentioned can be seen in the figure

-In vivo experiment is scarcely described, how many time authors treated the animals? How they deliver drugs? How they measured tumor volume along the time?

Response: I have add this information in the manuscript as L176 and L178-179 show

Without going any further, in my opinion this paper is to premature to be published.

Response: Thank you very much for your valuable advice.

We will further solve the various experimental problems you raised

In writing the article, we strictly follow the requirements of the magazine including experimental methods and picture format.

Round 2

Reviewer 4 Report

Unfortunately I have to note that the authors did not respond
properly to requests:

Response by authors: In this paper, the data were calculated. The control group was not a blank group, but a co ntrol group with the solvent. And the calculation formula is mean OD value of the experimental group/mean OD value of the control group×100%. Your data are not expressed in this way. Numerical example: control group after 72h has mean value of  0.8 OD

0.8/0.8*100= 100!

The authors completely ingnored the following experimental request of the referee...In addition, regarding the MTT method, authors should provide data also incubating cells with MTT salt for lower time; MTT is toxic for cells and they incubated cells with MTT for 72h, showing only one time point, presumably 72h. They should perform the same experiments incubating cells with MTT salt (0.5mg/ml) for only 2h at the end of treatments and they have to show also the effect on GBM viability at 24h (and 48h), that is the time point used for the scratch and transwell assay.

The authors completely ingnored the following  request...-please show the number of replicates: except for in vivo exps, authors never showed the N replicates per experiments, it’s incredible! Beyond the number of replicates, the figure legends lack of any basic information (treatment, time, concetration…)

Response by authors: I don't think it's impossible to do multiple controls and multiple trials with cells, and that's what we do. The problem you mentioned can be seen in the figure

I can't understand it.

Author Response

1. Unfortunately I have to note that the authors did not respond
properly to requests:

Response by authors: In this paper, the data were calculated. The control group was not a blank group, but a co ntrol group with the solvent. And the calculation formula is mean OD value of the experimental group/mean OD value of the control group×100%. Your data are not expressed in this way. Numerical example: control group after 72h has mean value of  0.8 OD

0.8/0.8*100= 100!

The authors completely ingnored the following experimental request of the referee...In addition, regarding the MTT method, authors should provide data also incubating cells with MTT salt for lower time; MTT is toxic for cells and they incubated cells with MTT for 72h, showing only one time point, presumably 72h. They should perform the same experiments incubating cells with MTT salt (0.5mg/ml) for only 2h at the end of treatments and they have to show also the effect on GBM viability at 24h (and 48h), that is the time point used for the scratch and transwell assay.

Response: The control group was used as blank to calculate the experimental data.

2. The authors completely ingnored the following  request...-please show the number of replicates: except for in vivo exps, authors never showed the N replicates per experiments, it’s incredible! Beyond the number of replicates, the figure legends lack of any basic information (treatment, time, concetration…)

Response by authors: I don't think it's impossible to do multiple controls and multiple trials with cells, and that's what we do. The problem you mentioned can be seen in the figure

I can't understand it.

Response: As for the problem of repeated experiments, we conducted multiple experiments to collect and process the data obtained from cells.The results show clear data from multiple experiments.The processing method, time and concentration can be found in articles and figures.